# E-Learning: Direct Effect of Student Learning Effectiveness and Engagement through Project-Based Learning, Team Cohesion, and Flipped Learning during the COVID-19 Pandemic

**Muhammad Umar and Ilsang Ko** *

Graduate School of Business Administration, Chonnam National University, Gwangju 61186, Korea;
umerstyle@gmail.com
* Correspondence: isko@chonnam.ac.kr

**Abstract:** The cross-impact of project-based learning, team cohesion, and flipped learning was investigated by examining their direct effects on student learning effectiveness, engagement, and engagement effects on learning effectiveness. The results of hypotheses testing were achieved using hierarchical regression analysis with SPSS-25 statistical packages for data analysis. The research model was empirically verified with quantitative data collected from 247 graduate/undergraduate business students based on their own experiences, observations, and engagement. The analysis found that project-based learning (PBL) and team cohesion increased positive direct effects both in student learning effectiveness and engagement. However, flipped learning showed increased positive direct effects in student learning effectiveness and negative effects on engagement. Furthermore, the engagement (itself) had a positive direct effect on student learning effectiveness. The proposed study was performed with the intention to inform practice in terms of increasing retention and enhancing teaching along with student learning quality.

**Keywords:** project-based learning; team cohesion; flipped learning; learning effectiveness; engagement

## 1. Introduction

The effective learning initiative is based on research by certain teachers and educators who have worked to guarantee that instructors may develop practices in ambitious directions so that all students, especially those who were previously overlooked by schools, can enjoy rich learning opportunities. Many concepts, such as "learning for knowledge", have been developed by educational scholars to describe learning and learning approaches that rely on students' understanding. While some of these concepts pertain to slight changes in approach, the majority of them imply that teachers should solicit, analyze, and investigate students' emerging ideas. In such a learning perspective, awareness is of the highest significance; it is actively negotiated by students and instructors, and it evolves as students' ideas arise and mature. The goal of this research is to examine the direct effects of student learning effectiveness and engagement. One major difficulty is determining how students practice and ensuring participation and engagement in light of unique learner qualities and digital interactions [1]. As we begin our creative approach, it is critical to identify different types of learning factors and their relationships with intermediaries. An interdisciplinary study of teacher–student relationships is a healthy connection for the benefits of this research.

The predictable outcome will be examined in this study as follows:

- How much do project-based learning, team cohesion, and flipped learning have direct effects on student learning effectiveness and engagement?
- How much does engagement directly affect student learning effectiveness?

These questions describe the student learning approaches and how they affect learning directions. The constructivist premise that learning is impacted by how an individual

participates in academically intentional activities underpins the concept of student learning. However, learning is viewed as a collaborative effort that requires institutions and personnel to provide students with the circumstances, expectations, and opportunities to participate. Individual learners are the ultimate participants in engagement dialogues. Through personal strengths, students may make significant contributions to the quality of a learning environment, their peers' experiences, and a wider campus community. Unfortunately, these potential benefits are not always fully realized. Fear plays an important part in the current pandemic situation, which distracts students from studying. Schools and universities have decreased their usage of online research during the pandemic, as well as social groupings, inadvertently preventing pupils from diverse backgrounds from engaging. Only a small number of students are meaningfully addressed by societies. Students' activeness and meaningful engagement are ignored or limited by established university governance systems. In this scenario, researchers are analyzing the direct effects on students whose learning was slower than it is now. Students can do their studies from home in the event of a pandemic, such as through flipped classes, project-based learning (PBL), and teamwork; this allows them to scientifically use technology, spend enough time debating the material, and have a thorough understanding of the themes covered. One of the most essential parts of the development of any country is the quality of learning in educational institutions. Therefore, educational institutions need to develop strategies that enhance the performance of the system [2]. Learning effectiveness is one of the crucial factors for advancing knowledge, innovation, modifications of teachers' techniques, and engagement [3]. The findings of this study will contribute to sharing practices within the academic community that improve learning style and student engagement. Teachers should work to enhance group learning by preparing lessons that are both fascinating and suitable for students to learn from [4]. This learning approach enhances the role of a student in engaging in a class and gaining knowledge through videos and related projects [5,6]. Learning is the basic aim, and teachers should have professional knowledge toward this aim. Maintaining a relaxed atmosphere can increase students' learning satisfaction [7]. Education is a replacement for orthodox teaching: it consists of obtaining information from videos and physical classroom meetings, encouraging students to exchange information, and fostering proactive awareness through interactive activities [8]. Teachers try to motivate students with kindness and acknowledge their efforts by engaging them in class. This teaching technique is an excellent way to make an unbiased and comprehensive assessment even though the primary purpose is to acquire knowledge [9]. Engaged students form relationships that are reported to affect their learning environments [10]. Study committed to the classroom experience, observation, engagement, ambition, outdoor activities, and the urge to participate in the learning process are all examples of student engagement with direct effects. It has been proven that being interactive with a student improves their attention and focus, motivates them to use better critical thinking skills, and develops valuable learning experiences. Students taught using a student-centered method of education show increased interest, which aids in the successful completion of the course's learning objectives.

## 2. Theoretical Background

### 2.1. Learning Effectiveness

Learning effectiveness is now focused on student learning; it is quite an active process, and can be formal or informal. It is a static group of reality, information, and skills. In effect, learning changes awareness, behaviors, abilities, morals, and preferences. Learning effectiveness refers to the entire process through which students participate in a high-quality learning opportunity. Within a system that fosters student growth, a high-quality education involves quantifiable results related to well-defined learning standards. The execution of strong academic objective targets made by a teacher and a group of instructors utilizing data about student learning over a defined period is referred to as student learning effectiveness. Evaluation leads to the growth of educational programs as well as the

assessment of their accomplishments and improvement of their effectiveness. Learning outcomes assist teachers and students in reaching a shared understanding regarding the aims and objectives of a course or an academic program. Learning effectiveness refers to the entire process through which students participate in a high-quality learning experience. Specific design features such as cognitive, instructional, and social representation in the e-learning network have been used to assess learning efficacy. Through thought and research, these design components aim to cultivate and facilitate higher-level thinking skills. A question that learning effectiveness addresses is: "Do I understand more than I knew before? How will this new knowledge help me?" Effective learning is the capacity of a learner to explicitly indicate what they have learned through quantitative measures. Learning begins with the formulation of particular learning outcomes that are aligned with a course's or program's general goals and objectives. In this study, the outcomes of learning are related to direct effects on learning through project-based learning, team cohesion, and flipped learning. The students' learning with regard to effectiveness, actual accomplishment, morale, and knowledge, and the team's capacity to build and maintain an excellent learning atmosphere are all defined and studied. According to experts, e-learning represents a gateway to development and growth for emerging countries. According to the authors of [11], the scope of expectations and validity for emerging and developed countries differs substantially. In a previous study, the effects of learning on the deployment of student learning were studied [12]. The Table 1 shows that the research construct and prior research variables outcomes of the different authors in learning effectiveness.

**Table 1.** Key findings of research.

| Research Constructs | Research Variables | Prior Research Variables and References |
|---|---|---|
| Learning effectiveness | | Teacher's effectiveness, teacher's experience, teacher's professional knowledge, professional development, teacher's content knowledge [13]. Online tools and resources, interactions, technology quality, self-regulation, attitudes towards, blended learning, motivation, satisfaction, knowledge construction [14]. E-learning effectiveness [15]. Effectiveness, performance, self-efficacy, satisfaction, human dimension, student, instructor, design dimension, learning model, technology, learner control, content, interaction [16]. |
| Project-based learning (PBL) | Critical thinking | Critical thinking, institutional growth in critical thinking, institutional selectivity [17]. Project-based learning (PBL), critical thinking [18]. Project-based learning (PBL), critical thinking, internal influences, external influences, beliefs about projects, tools for technology, learning outcomes and products [19]. |
| | Communication | Project-based learning (PBL), communication skills, essential question, research and writing, product creation, presentation, evaluation and reflection [20]. Communication, self-perceptions, social competence [21]. Learning soft skill, oriented innovation [22]. |
| | Collaboration | Student–teacher practice, co-teaching experience, co-teaching interactions [23]. Collaborative, knowledge outcome, skill outcome [24]. Collaborative, experimental group, control group [25]. |
| | Creativity | Innovative technologies, students' creative activities, creative ability execution/translation, conditions for co-creativeness [26]. Creativity, subject-specific questions, decontextualized questions [27]. Creative resources, individual-team level, knowledge, and performance, behavioral [28]. |

**Table 1.** *Cont.*

| Research Constructs | Research Variables | Prior Research Variables and References |
|---|---|---|
| Team cohesion | Individual trust | Trust, cohesiveness, performance, blended service, effective co-production [29].<br>Institution-based trust, swift trust, virtual team trust, trust-building skills, deterrence-based trust [30].<br>Trust in leaders, trust in team members, collective efficacy, teamperformance [31].<br>Propensity to trust, perceived trustworthiness, cooperative behaviors, monitoring behaviors, perceived task performance, team satisfaction, attitudinal commitment, continuance commitment [32]. |
| | Commitment | Sense of control, positive emotions, perceived cohesion, commitment behavior [33].<br>Team behavior, task conditions, intended effects [3]. |
| | Responsibility reporting | Teamwork, team adaptation, interpersonal interaction [34].<br>Sustainability reporting, size, ownership, industry [35]. |
| | Effective coordination | Transition adaptation, reacquisition adaptation [36].<br>Coordination behavior, clinical performance, leadership experience, team size, duration of the scenario [37].<br>Team processes, transition processes, action processes, interpersonal processes [38]. |
| Flipped learning | Peer and teacher interaction | Teacher–student interaction [39].<br>Teachers–students [40].<br>Interaction, peer interaction [41]. |
| | Flexible atmosphere | Learning environment [42].<br>Learning environment, online education, adaptation of curriculum [43].<br>Learning environment, agentic engagement, motivational support [44]. |
| | Learning culture | Perceived learning culture, developmental feedback, interaction effects, team creativity [45].<br>Traditional culture, transformational culture, culture on student personality [46].<br>Theory of intercultural, intercultural education [47]. |
| | Problem-solving activities | Problem-solving skills, flipped classroom, planning, evaluate, expect [48].<br>Problem-solving skills, problem-solving difficulties [49].<br>Problem-solving abilities, classroom interaction [50]. |
| | Professional educators | Teacher professional development, professional teaching, digital tools [51].<br>Professional growth, traditional/reform professional development, teachers' achievements [52].<br>Professional instructor, online preparation, face-to-face, follow-up [53]. |

**Table 1.** *Cont.*

| Research Constructs | Research Variables | Prior Research Variables and References |
|---|---|---|
| Engagement | | Student engagement, online learning [54]. Behavioral, cognitive and emotional engagement [55]. Academic commercialization and technology transfer, knowledge exchange involves academics, improving innovation and business performance [56]. Motivation and agency, transactional engagement, institutional support, active citizenship, social beliefs and practices [57]. Student motivation, transactional engagement, transactional engagement, institutional support, active citizenship, non-institutional support [58]. Student engagement, interaction, assessment for learning, instruction, multimedia, engaging and challenging technology, relevancy, exploration [59]. Transition engagement scale, academic engagement scale, peer engagement scale, student–staff engagement scale, online engagement scale, beyond-class engagement scale [60]. Student engagement, academic and financial aid information [61]. Student engagement, student learning [62]. Student engagement and achievement, student-reported engagement, teacher-reported engagement [63]. Behavioral, emotional, cognitive engagement [64]. |

## 2.2. Project-Based Learning

One of the most popular symbolic applications of virtual videos in project-based education is Khan Academy. The idea is that all learners in one area may use the same materials and sources; they can arrive early and work at their own pace. Engaging in different activities dominates students' roles [65], but face-to-face interaction is the most valuable for students' teaching as a technique to engage students in learning and project-based learning (PBL) [66]. As is evident in a previous study [67], when the authors used a project-based learning method, the professional identity of students and teachers was developed. A study guide to real-world projects in the digital era, reinventing project-based learning (PBL), was discussed [68]. They found different information and learned to customize facts and tools to accomplish learning objectives, in addition to getting comfortable in the student role as an architect, expressing intentions, and determining what effort should be put into achieving key learning goals. The study construct and prior research variables findings of the different authors in project-based learning are shown in Table 1.

### 2.2.1. Critical Thinking

Critical thinking provides a final perspective on a project's ability to be finished on time. Students' learning activities, considering the beliefs they hold, can help provide learning techniques, resources, and personalized assistance. The ability of students to think deeply reflects the differences between analytical and impulsive cognitive types. According to [17,18] through critical thinking, enhanced institutional growth, and institutional selectivity, students can improve their capacity to think rationally by incorporating these aspects into the learning process [19]. The authors assert that critical thinking in project learning is internally and externally influenced; students can hold beliefs about projects, tools for technology, learning outcomes, and products. This study demonstrates that advanced critical thinking skills include the capacity to objectively think about events and misconceptions, focus on techniques, and locate information relevant to actual settings. Determining students' opinions is vital in this era of the digital world or e-learning to learn about beneficial thoughts and sentiments of any given entity.

Table 1 shows the study concept and prior research variables findings of many scholars in critical thinking.

### 2.2.2. Communication

If students want to participate, they must communicate. Foster learning enhances student communication by assisting them in reaching their goals, expanding possibilities, strengthening the student–teacher connection, and providing a pleasant overall experience. In [20], for the author, communication in project-based learning demonstrates the team members' mutual understanding and likeness. Students' communication abilities affect their abilities in asking important questions, research and writing, product creation, presentation, assessment, and reflection. Likenesses in communication may reflect students' self-perceptions and social competence [21]. With regard to communication skills, soft skills learned by students are important for oriented creativity [22]. According to this study, students participate in group learning with their classmates, which improves the learning experience. Some academics suggest that communication learning is a practical training strategy since it helps students develop their capacity to communicate vocally and in writing while also allowing them to apply what they have learned in class to their everyday lives. The majority of the findings reported in the literature are positive. Table 1 shows the study construct and prior research variables findings of the many authors in communication.

### 2.2.3. Collaboration

Collaborative learning can help students improve their higher-level thinking abilities, spoken communication skills, and self-management, among other things. In [23], the author claimed that student–teacher practice, the co-teaching experience, and co-teaching relationships enhanced the good moral experience. Increased knowledge and abilities obtained generate good feedback when teachers and students collaborate [24]. Both the experimental group and control group examined the act of student learning in a group of people that worked together [25]. Table 1 shows the research construct and prior research variables findings of the many scholars in collaboration.

### 2.2.4. Creativity

In this study, creativity is explained via the research of invention and imagination. Creativity increasingly fosters imagination to prepare pupils for the future. The utilization of innovative technologies boosts the creative potential of students [26]. It must be included into the educational process; in other words, it cannot be taught as a separate subject that prepares pupils for the future [27]. The primary issue with student creativity is that it uses resources in a long-term manner [28]. The educational response to the need to develop the imagination required for a student's potential success is creativity. An overview of opinions described in the literature is shown in Table 1. The findings of the many authors in creativity on the study construct and prior research variables.

### *2.3. Team Cohesion*

Team cohesion allows for the interpersonal links between team members of a group to be strengthened. Cohesion within a team is important for student learning since it leads to higher performance, enhanced student satisfaction, and increased student motivation. It does not matter what kind of dynamics exist in the team. All team members must know their particular job and purpose and be confident that each person contributes to the effort [69]. Student sports classes encourage cohesion and learning satisfaction. Some students are very social, some are not social, and some are selfish, making for a better or worse team combination [70], individual qualities and team outcomes are linked through a complex mechanism. Changes in wealth or any apparent differences in performance among groups should not be a reliable measure of the outcome of cohesion. Variation in team cohesion may be attributed to the effect that these differences have on team members' desire

to participate in creative activities [71]. Furthermore, the character of a team is probably more relevant than individual style, given that a team's growth does not provide personal success. It is not true that team cohesion depends on team participants' liking for each other; it is affected by whether attention is paid to all team members, whether questions are asked, and whether knowledge is shared, which have all been analyzed in studies and fieldwork, whether in or out of class. Teamwork improves team efficiency, and fewer time-consuming activities can help motivate team members. This research investigated how teams of business students differed in team unity, team member satisfaction, presentation, and project engagement plans in relation to student learning, social cohesiveness, task cohesion, and team performance [72]. An overview of the literature Table 1. Shown that team cohesiveness, the study construct and prior research variables outcomes of the many researchers.

### 2.3.1. Individual Trust

A team with reliability and integrity will succeed, but any person in the group will benefit from the presence of integrity since they are part of a vital, cohesive community. [29]. A class or project will achieve concrete goals when individuals trust each other, according to the author of [30], who claims there are various types of trust: institution-based trust, rapid trust, virtual team trust, trust-building skills, and deterrence-based trust. [31] According to the author, if students have faith in the leader, they will work together as a team. On the contrary, trustworthiness and cooperative actions dominate students' character building [32]. Table 1 shows that the points of view based on research constructs and prior research variables.

### 2.3.2. Commitment

When all team members agree to and decide on a single strategy, commitment to team priorities is produced [3]. Project liability refers to teammates' shared duty to increase their joint efforts in completing each task. When people work together with their choice, going through disagreements, coordinating with each other, playing together, facing challenges, and holding each other to high standards, each team player focuses on achieving the team's goal beyond their individual goals [33]. One element required for people to feel strong desire or commitment is to meet and carry out learning engagement on a project or an individual basis. An overview of perspectives published in the prior research is shown in Table 1.

### 2.3.3. Responsibility of Reporting

The duty to report is frequently described as a cooperative effort among project team members. Reporting responsibility is commonly defined as a group of people's desire to collaborate and engage with one another to achieve a common goal [34]. Given that many of their duties can be hampered by impediments in the hands of teams, it is up to students to explain the process and grasp what would improve learning, if not perfect it, within sustainability reporting [35]. We focus on students' reporting and actions with their peers in a team endeavor during this phase. Members are more likely to participate and stay enthusiastic to satisfy the reporting criterion because of this psychological tie. The research notion and prior research variables findings of numerous researchers in charge of reporting are shown in Table 1.

### 2.3.4. Effective Coordination

This research looks at ways to get students to cooperate in class and how to include them in practical learning. In our research, the group influence of trust and team unity equates to successful and effective coordination. In [36], according to the authors, transition adaptation and reacquisition adaptation successfully create a positive feedback loop for overall project performance. Coordination allows team members to work together to change and improve their team dynamics within a project, experience, performance,

and team size [37], which is a great strategy to increase collaboration and ensure team learning outcomes. Interpersonal processes, action processes, and action processes are all made possible by successful coordination [38]. The research constructs and prior research variables findings of many researchers are shown in Table 1 in a well-coordinated manner.

*2.4. Flipped Learning*

The benefits of a flipped classroom are important for both instructors and students. One of these is that it encourages students to shift away from traditional learning. It brings them closer to proactive learning skills, in which teachers and students may actively collaborate, combining a flipped classroom with project-based group work [41]. According to the author of [73], learners study online courses, and their questions are responded to by the instructor throughout the time in class. This study shares the ideas suggested using flow learning dimensions. While they do not understand the effect this method has on group results, task-wise, studies have been restricted to accurately handling problematic situations to improve learning challenges [74]. The flipped classroom encourages higher-order thinking; for example, speaking ability was improved in an EFL flipped classroom, enhancing student engagement, and flipped learning was used to improve the teaching of ICT engineering. The implementation of a flipped classroom also improved students' social interaction and critical thinking skills [45,75–78]. The research construct and prior research variables findings of the many writers in flipped learning are shown in Table 1.

2.4.1. Peer and Teacher Interaction

When conducting any task or training, teachers create a specific activity in peer interaction, which is described as the connections and roles of the students in the classroom context. Whole-class, solo, pair, and group work are frequently established by teachers as key components for students to exchange and develop their knowledge about the course and class activities [21]. In this study, student–teacher interaction in a flipped classroom is a key component of this relationship. In [39], according to the author, in the classroom, students listen to one another's inquiries and create rapport via regular contact, which is beneficial to both students and their peers. In this way, our grasp of the core qualities of peer contact, its impact on student progress, and its educational potential lag far behind our understanding of the interaction between teachers and students. According to Albert Bandura's social learning theory, the observation of interaction with peers and teachers enables us to understand social and emotional learning. This aids in the learning, comprehension, and strengthening of peer connections; the study in [40] reveals that students connect with their peers and professors when they educate other students by design and receive assistance from their peers. The consequences might be either beneficial or harmful. Peers' negative influence on a student's academic achievement is distressing. The research construct and prior research variables outcomes of the different authors in peer and teacher interaction are shown in Table 1.

2.4.2. Flexible Atmosphere

The classroom learning technique is linked to a comfortable learning environment, and teachers must have a broad range of expertise and a more holistic approach. In this study, we focus on how to utilize students in the learning environment in the classroom and group environment. In [42], the authors conclude that videos take up the majority of class time and that proponents of the atmosphere technique, in which a teacher employs unlimited class time, are prominent. In [43], the author claimed that, compared to in regular classrooms, pupils spent much more time in a class and significantly more time in groups due to the restrictive environment. In general, students in a relaxed environment spend a greater proportion of class time actively engaged in the class lesson [44]. Table 1 shows the study framework and prior research variables outputs of the flexible environment by different authors.

### 2.4.3. Learning Culture

Culture gives you opportunities to explore new thoughts and visions. Understanding how community learning influences student learning with cultural characteristics has a substantial impact on the learning process [46]. In this research, we look at how to create a meaningful environment in which everyone feels at ease and can contribute their expertise. Learning cultures are present in the classroom, are shaped through teaching and learning activities, and are centered on students' personal development. According to the author of [47], community contributions include social adjustment elements, transformative cultures, and the effect of cultures on student personality, which means student engagement plays an important part in students' learning. The impact of culture on a particular learning context should be acknowledged, and the culture should be altered as much as possible to encourage the type of learning environment that has been shown to have favorable outcomes [48]. Table 1 shows the learning culture research construct and prior research variables outcomes of various authors.

### 2.4.4. Problem-Solving Activities

Problem-solving exercises are an excellent way to discover how team members work individually and in groups. It is vital to teach pupils how to quickly overcome obstacles that stand in the way of any undertaking, such as through planning, evaluating, and problem-solving skills [49]. Problem-solving is influenced by these elements, which include aims, memory, attention, and perception [50]. These are higher cognitive processes used to solve a problem and achieve a specific goal; they differ depending on the information, experience, and talents required to solve the problem [51]. The problem-solving activities research construct and prior research variables outputs of different authors are shown in Table 1.

### 2.4.5. Professional Educators

Diverse teaching, student issues, and diversified learning are all best handled by an experienced teacher. Many issues have been documented in the literature regarding traditional and online professional educators; they use professional teaching, technological devices, online preparation, and face-to-face contact [52]. Both the instructor's professionalism and the course's technique enabled the problems of students' participation to be solved in this study. Professional teaching is widely acknowledged to include the possession of abilities, practices, and knowledge that influence the professional growth and development of one's teaching career [53]. Digital tools should be used to support teachers' professional development more consistently [79]. The influence of professional learning on the learner is critical [54]. This strategy swaps EFL instructors in the classroom and homework chores to promote active learning, engagement, and accomplishment. Table 1 shows the professional educators research construct as well as prior research variables from other authors.

### 2.5. Engagement

According to [80], learning about thinking is called a cognitive process. Engagement can be the rate at which learners are captivated by their instructive activities [81]. The previous findings show that children think differently about academic tasks and that rethinking the traditional learning pattern is critical. The combination of formal and informal methods to deliver learning assistance appears to have worked successfully. However, building human relationships with students was among the problems, making it difficult to engage the cohort in learning [82]. Table 1 shows the outcomes of many authors' engagement research constructs and prior research variables.

### 3. The Research Model and Hypotheses

Built on the theoretical background introduced in the previous section, a research model was proposed to investigate the effect of student learning effectiveness and engage-

ment, shown in Figure 1 [64]; learning effectiveness is one of the most important aspects in expanding knowledge, creativity, teacher method changes, and student engagement.

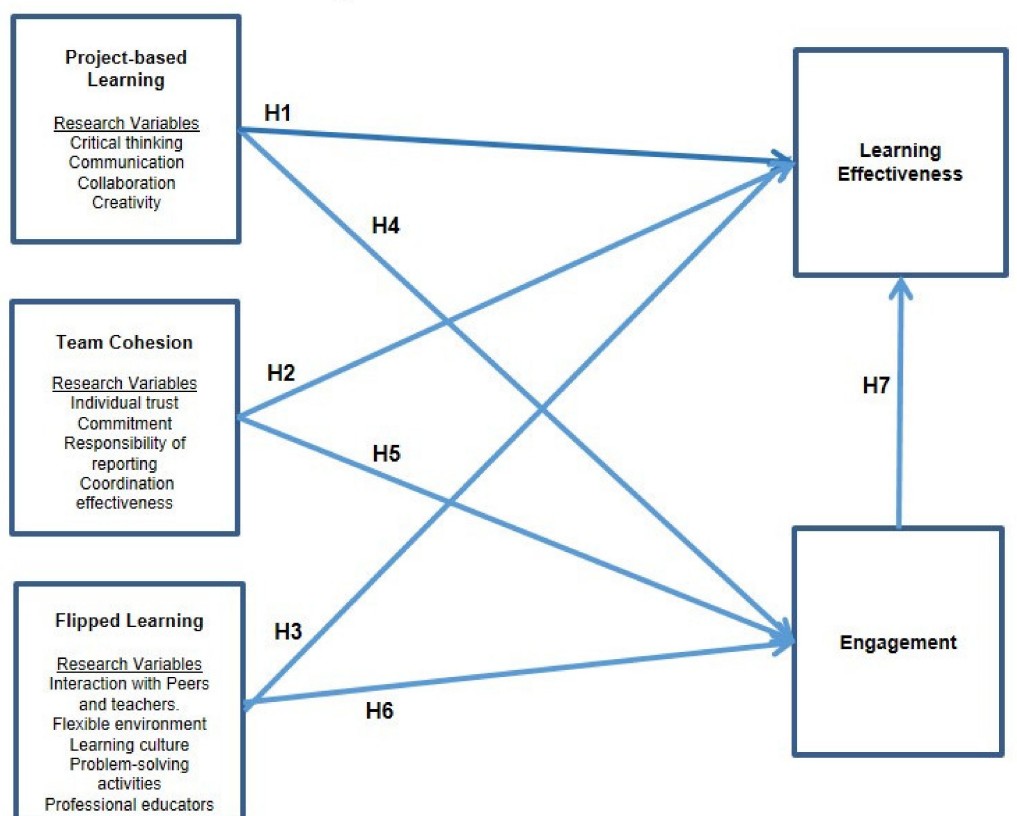

**Figure 1.** Conceptual research model.

*3.1. Hypotheses*

**Hypothesis 1 (H1).** *Project-based learning (PBL) increases student learning effectiveness.*

**Hypothesis 2 (H2).** *Team cohesion increases student learning effectiveness.*

**Hypothesis 3 (H3).** *Flipped learning increases student learning effectiveness.*

**Hypothesis 4 (H4).** *Project-based learning (PBL) increases engagement.*

**Hypothesis 5 (H5).** *Team cohesion increases engagement.*

**Hypothesis 6 (H6).** *Flipped learning increases engagement.*

**Hypothesis 7 (H7).** *Engagement increase student learning effectiveness.*

The two approaches employed are direct assessments of student learning effectiveness and engagement [83]. Best practices employing direct evaluations to measure the degree of student learning.

*3.2. Direct Measures of Student Learning Effectiveness*

In this study, direct measures were used to assess the model, which directly affected student learning effectiveness. This study addresses three major factors of student learning:

(1) project-based learning, (2) flipped learning, and (3) team cohesion. The study extended the value of learning by evaluating a class as described in the approach and providing a valuable opinion that the outcomes of online learning are at least minimally equivalent to those of other modes of delivery. Learning effectiveness, such as the relevance of knowledge, focuses on different variances in a learner's understanding, talents, and ways of learning [84]. Direct learning is defined in this study as individuals', groups', and team members' fulfillment, real accomplishment, morale, knowledge communication, and the team's capability to produce great learning. A decade of research suggests that the flipped class delivers excellent academic effects [85,86]. Students' learning performance and confidence can be improved by flipping the classroom [87]. Through knowledge sharing, students may participate in and interact with the model with the support of peer-assisted learning. Using the flipped class as a teaching method and attitude forces both teachers and students to evaluate the teacher's teaching and students' related outcomes [88]. Project-based learning (PBL) has been shown to foster a "need to know" mentality in students by arranging learning around relevant direct goals. Students are motivated to expand their knowledge to address an issue that is significant to them. There has been substantial growth in the prevalence of project-based learning (PBL) and serious concerns regarding its arrival. Critics of project-based learning (PBL) question whether the emphasis on practice helps teachers to use a technical education technique rather than to promote teaching sensitive to the ideas of students. The project-based learning (PBL) technique is therefore highly recommended for students' usage in education and should be supported in institutions. As a result, the project-based learning (PBL) technique provided a sense of connection with the content for both the course and its classmates. The cohesiveness of teams relates to how dedicated teams are to their goals and how effectively the teams are integrated into achieving those goals directly [89]. Team members have a robust sense of belonging and are more inclined to collaborate, engage, and exchange ideas with each other [90]. Therefore, its impact on team members' incentives to undertake creative work is partly due to the benefits of team cohesiveness. Direct measurements are preferred for assessing student learning levels on specific goals since they represent what students can perform.

### 3.3. Direct Measures of Student Engagement

In this study on learning, direct metrics were employed to analyze the model, which has a direct influence on the efficacy of student engagement. This research looks into three key aspects of student learning: (1) project-based learning; (2) team cohesiveness; and (3) flipped learning. According to [91], the use of a flipped class to improve student engrossment encouraged active learning even outside the classroom [92]; project learning activities based on real-world events might be used in class to increase students' understanding and comprehension of the material while also encouraging them to articulate their involvement in such activities [93]. As a result of the learning environment, the deployment of the flipped class was shown to improve student involvement and result in favorable learning results, improving their performance. The findings indicate that collaborative understanding is more likely to be retained and that students can gain a deeper understanding of various subjects indirectly [94,95]. Students promote more positive attitudes toward project-based learning than any other type of learning, whether direct or indirect instruction is used. This is agreed with by the authors of [96], who examined the studies on PBL and found evidence that students using collaborative PBL had academic achievements similar to or superior to the achievements of students using different methods of learning. The cohesiveness of a team is determined by how committed its members are to it and how well they work together to achieve its objectives. It affects team members' motivation to do creative work, which is partially related to the advantages of team cohesion. Although there is scientific evidence of a link between team cohesion and creativity, cohesive teams do not create creativity on their own. Significant levels of team cohesion may contribute to a decrease in performance via group thinking and conforming processes [96]. Learners' incapacity to maintain their online education has been attributed to a lack of support or

an increase in workload, resulting in learner attrition [59]. The relevance of knowledge, for example, focuses on distinct variations in a learner's comprehension, abilities, and learning styles [50]. By assessing a class as stated in the method used in [51], we show that the effects of online learning are minimally equal to those of other teaching methods. Individuals', groups', and team members' fulfillment, actual accomplishment, morale, knowledge communication, and the team's capacity to establish and sustain a high level of learning effectiveness are all characterized as learning in this study. Students commit effort and time to their education and conduct, for which an organization gives learning opportunities and facilities [60]. Many researchers relied on students' engagement, passion, and commitment to social and educational activities to encourage greatness and performance improvement [61]. Students can gain a lot from their involvement [62]. To these degrees, it enhances student focus, communication skills, expressiveness, and the experience and understanding of thinking. To measure student learning levels on specific goals, indirect measurements are recommended since they specify what students can do.

## 4. Data Collection and Statistical Analysis

In this research, the supervisor reviewed the survey questionnaire, determined the information needed, decided on a question topic, and developed question content. The questionnaire's criterion was that respondents understood the question's objective. Before starting the survey, each class had an introduction to the questionnaire and the purpose of the survey. A questionnaire is a type of quantitative research tool that consists of many questions that are used to collect data from respondents. In total, 250 survey copies were distributed; 3 questionnaires were discarded because respondents provided incomplete survey information. Two hundred and forty-seven effective samples of the questionnaire were completed and returned, with a survey return rate of 98.8%. In-person feedback was used to acquire data for the survey.

The Statistical Product software package SPSS 25 was employed to calculate the statistics concerning data processing. A data-gathering survey was conducted with a specific set of business management students. The survey was carried out among undergraduate and graduate business management students at Chonnam National University in various programs/courses in the school of business. Participants in the poll came from different countries and were enrolled in courses at Chonnam National University.

### 4.1. Scales

In this study, all measurement construct extents used a five-point Likert Scale from strongly disagree to agree strongly with the scale from 1 to 5, as follows:

Strongly Disagree (1), Disagree (2), Neutral (3), Agree (4), Strongly Agree (5).

### 4.2. Gender

In Table 2, we summarize the participants. The number of participants was two hundred and forty-seven.

**Table 2.** Gender frequency and percent.

|  |  | Frequency | Percent |
|---|---|---|---|
|  | F | 116 | 47% |
| Valid | M | 131 | 53% |
|  | Total | 247 | 100 |

### 4.3. Participants Country-Wise

As shown in Table 3, country-wise, most of the participants were from South Korea, Uzbekistan, China, Pakistan, and other neighboring countries in Asia, Europe, and Africa.

**Table 3.** Country-wise participant frequency.

| Countries | Frequency | Percent | Valid Percent | Cumulative Percent |
|---|---|---|---|---|
| China | 9 | 3.6 | 3.6 | 3.6 |
| Croatia | 1 | 0.4 | 0.4 | 4.0 |
| Dominican Rep | 1 | 0.4 | 0.4 | 4.5 |
| East Timur | 3 | 1.2 | 1.2 | 5.7 |
| Ethiopia | 1 | 0.4 | 0.4 | 6.1 |
| France | 4 | 1.6 | 1.6 | 7.7 |
| Iran | 1 | 0.4 | 0.4 | 8.1 |
| Kazakhstan | 7 | 2.8 | 2.8 | 10.9 |
| Luxemburg | 1 | 0.4 | 0.4 | 11.3 |
| Malaysia | 1 | 0.4 | 0.4 | 11.7 |
| Pakistan | 9 | 3.6 | 3.6 | 15.4 |
| Russia | 3 | 1.2 | 1.2 | 16.6 |
| South Korea | 137 | 55.5 | 55.5 | 72.1 |
| Uzbekistan | 68 | 27.5 | 27.5 | 99.6 |
| Vietnam | 1 | 0.4 | 0.4 | 100.0 |
| Total | 247 | 100.0 | 100.0 | |

*4.4. Descriptive Statistics*

The initial step in our analysis was to compile descriptive statistics for the variables in our research. Project-based learning (PBL) ranged from 1.00 to 5.00, with a mean of 3.87 and a standard deviation of 0.916, according to the descriptive statistics for our sample. With varying means and standard deviations, team cohesion, flipped learning, engagement, and learning effectiveness ranged from 1.00 to 5.00. (See Table 4).

**Table 4.** Descriptive statistics.

| | N | Minimum | Maximum | Mean | Std. Deviation |
|---|---|---|---|---|---|
| PBL | 247 | 1 | 5 | 3.87 | 0.916 |
| TC | 247 | 1 | 5 | 3.64 | 0.819 |
| FL | 247 | 1 | 5 | 4.18 | 0.749 |
| Engagement | 247 | 1 | 5 | 3.81 | 0.721 |
| Learning Effectiveness | 247 | 1 | 5 | 3.65 | 0.802 |
| Valid N (listwise) | 247 | | | | |

*4.5. Model Summary*

Based on the findings of the measurement model evaluation, we used hierarchical regression analysis with OLS to assess our research hypotheses. The significant and non-significant coefficients in the study model are reported in Table 5, and the results confirmed the research hypotheses. In our first model, direct effects of learning effectiveness revealed that project-based learning had a marginally positive effect ($\beta = 0.025$, $p = 0.06 > 0.05$), team cohesion had a significant, positive effect ($\beta = 0.948$, $p = 0.000 > 0.01$), and flipped learning had a marginally positive but non-significant effect on student learning effectiveness ($\beta = 0.019$, $p = 0.238 > 0.1$). In our second model, the direct effects of engagement revealed that project-based learning had a positive, significant effect ($\beta = 0.113$, $p = 0.023 > 0.05$),

team cohesion had a significant, positive effect (β = 0.111, *p* = 0.046 >0.05), flipped learning had a negative but significant effect on student engagement (β = −0.143, *p* = 0.018 > 0.1). In model three, engagement had a positive but not significant effects on student learning effectiveness (= 0.128, *p* = 0.070 > 0.05).

**Table 5.** The results of hypotheses testing using hierarchical regression analysis with OLS.

| Variables | Learning Effectiveness | Engagement | Learning Effectiveness |
|---|---|---|---|
| | Model 1 | Model 2 | Model 3 |
| PBL | 0.025 ** | 0.113 ** | - |
| TC | 0.948 *** | 0.111 ** | - |
| FL | 0.019 | (0.143) *** | - |
| Engagement | - | - | 0.128 ** |
| $R^2$ | 0.947 | 0.066 | 0.013 |
| *Adjusted $R^2$* | 0.946 | 0.055 | 0.009 |

Note: *** *p* < 0.01; ** *p* < 0.05; * *p* < 0.1, model 1: dependent variable = learning effectiveness; model 2: dependent variable = engagement; model 3: dependent variable = learning effectiveness.

## 5. Research Results

*Hypotheses Analysis*

Using hierarchical regression analysis, we discriminated between study variables to establish the direct relationship. We established hypotheses based on our conceptual research approach in this study. As shown in Table 6, (H1) project-based learning (PBL) has a positive but marginally significant influence on student learning effectiveness. As a result, we accepted hypothesis (H1). We accepted hypothesis (H2) that team cohesiveness has a significant positive influence on students' learning effectiveness. (H3) Flipped learning has a good effect on student learning effectiveness, although it is not statistically significant. As a result, hypothesis (H3) was rejected. (H4) Project-based learning (PBL) and (H5) team cohesiveness have significantly positive effects on student engagement. Therefore, we accepted both hypotheses (H4) and (H5). (H6) Flipped learning negatively affects engagement but significantly supports student engagement. Therefore, we accepted hypothesis (H6). Lastly, (H7) engagement has a positive effect on student learning effectiveness but is not significantly supportive of student learning effectiveness; therefore, hypothesis (H7) was not accepted. Figure 2 shows the model summary results and shows the strength of the relationship between the research constructs and dependent variables.

**Table 6.** The results of hypotheses testing with hierarchical regression analysis.

| Hypotheses | Variables Paths | Unstandardized Coefficients | Standardized β Coefficients | S.E. | T | *p*-Value | Accepted or Rejected |
|---|---|---|---|---|---|---|---|
| H1 | PBL→LE | 0.025 | 0.028 | 0.013 | 1.879 | 0.061 | Accepted |
| H2 | TC→LE | 0.948 | 0.968 | 0.015 | 64.574 | 0.000 | Accepted |
| H3 | FL→LE | 0.019 | 0.018 | 0.016 | 1.183 | 0.238 | Rejected |
| H4 | PBL→Eng | 0.113 | 0.144 | 0.049 | 2.286 | 0.023 | Accepted |
| H5 | TC→Eng | 0.111 | 0.126 | 0.055 | 2.008 | 0.046 | Accepted |
| H6 | FL→Eng | −0.143 | −0.148 | 0.06 | −2.389 | 0.018 | Accepted |
| H7 | Eng→LE | 0.128 | 0.115 | 0.071 | 1.820 | 0.070 | Rejected |

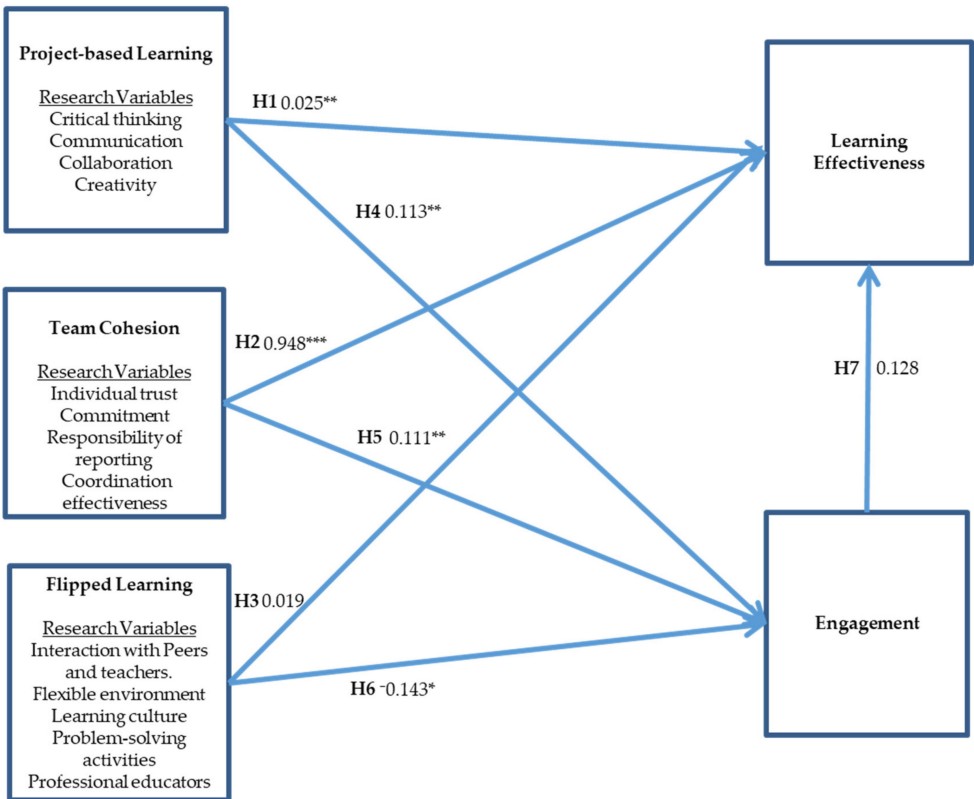

**Figure 2.** Results of the research model. Note: *** $p < 0.01$; ** $p < 0.05$; * $p < 0.1$.

## 6. Conclusions and Discussion

In our research, we discovered that project-based learning (PBL), team cohesiveness, and flipped learning had direct effects on student learning effectiveness and engagement. There is not only a statistically significant, positive relationship, but also, there is a statistically negative relationship. This study showed that project-based learning (PBL) had a beneficial impact on student learning effectiveness and engagement, as predicted. Several studies [18,20,25,28] have mentioned research variables in project-based learning (PBL). The results were also compared to other studies, which indicated the parameters had a substantial impact on student learning effectiveness [20,23]. On the other hand, the findings revealed that team cohesion plays a dominant role in a student's learning effectiveness and engagement. Although research variables were counted in receptive studies [30,33,34], the results were found to be significantly similar to those of studies that claimed the factors were important in influencing student learning effectiveness, as shown in [33,34] and [36]. Another noteworthy observation is that while flipped learning had a statistically significant beneficial effect on student learning effectiveness, it also had an unanticipated negative effect on engagement. In [41,47,51,80], it was claimed that flipped learning has little effect on student learning. The results were also compared to those of previous studies that affirmed the importance of various elements in determining student learning effectiveness [46,50,51]. Finally, while engagement has a direct impact on student learning effectiveness, it is not well substantiated.

During the COVID pandemic, most of the students enrolled in online coursework. The impact of this type of study at the university level for local and international students can be learned from future research and advancements. This study involved undergraduates who had appropriate student engagement, their class or group experiences, and observations. Students prefer active learning environments over passive classroom settings, and they expect the same in their classrooms because they live in such an engaging society. Today's students want more opportunities for creativity and collaboration, which virtual education environments may deliver through a range of teaching methods that can be

accessed at any time and from any location [44]. Previous studies have shown that students' participation helps them to achieve academic improvement and create a more motivating learning environment.

## 7. Contribution

In this study, project-based learning, team cohesiveness, and flipped learning all had significant roles in student learning effectiveness and student engagement. These findings imply that interacting with students and instructional material that is updated regularly might improve students' perceptions of learning. The majority of nations are dealing with the COVID-19 epidemic, and most have switched to digital or virtual education systems. Furthermore, because university education has substantial socioeconomic repercussions [97], this is a great example of a "case study including students and their contributions" to increase learning efficacy. One case study aimed to look at the idea of an ideal cow ranch from the views of two different groups [97]. A group of young people was motivated to propose a research method that would encourage students to reflect on and assess their knowledge in comparison to that of others. When learning patterns for non-traditional and traditional students were taken into account, only a tiny fraction of students employed terminology that showed a logical approach to dairy farming, such as ethical concerns, liberty, sustainability, wildlife distress, and organic dairy production. According to the findings of the case study, a curriculum that will help young people become more aware of current concerns in dairy production should be developed. To learn more successfully during the epidemic, training and online classes might be a preferable alternative. As a result, governments are concerned about ensuring a continuous flow of students enrolling in higher education. According to our findings, the digital education system is influenced by three major variables. In higher education, flipped teaching has several advantages. It enables students to study, comprehend, and interact with classroom lectures, as well as take ownership and responsibility for their learning. As a result, they play an important role in academic society. However, according to this study, project-based learning in academic contribution enhances a student's ability to work with others, as well as cooperation and group capacities to share an idea in a group project and engage in positive roles in society. Project-based learning improves an instructor's understanding of a student's circumstances. An investigation into team cohesiveness revealed that the strength and range of interpersonal relationships among members of a group are characterized as team cohesiveness. Members of academic societies are more likely to participate and stay motivated to achieve the stated goals as a result of this human connection.

## 8. Practical Implications and Limitation

This study was a beneficial tool in identifying how to measure the effectiveness of learning studies. Methodological research must continue to account for measurement error when studying self-performance and can add other aspects to the mix, such as game-based learning and cooperative learning. Students who are unfamiliar, perplexed, unwelcoming, or unsupportive in the classroom will struggle. In other institutions, other relevant aspects that influence the effectiveness of student learning can be included. While the authors of [98], contend that the earlier study's reaction to the input explanation was quite low, and while idlers are common in empirical investigations and may reduce the comparability of assumptions, a new study is needed to determine the best framework for analyzing student learning effectiveness. This study examined the consequences of a developing consensus on student learning and effectiveness for students and classroom methods, as well as a modern mix of elements that connect digital learning. As a result of their learning experiences, students experience anxiety, frustration, social isolation, sadness, and a lack of self-confidence. Foreign students studying abroad, in particular, experience language barriers, differences in cultural norms, value issues, and difficulties in different education systems. The results of this study may assist learners by providing comfort, healthy growth, and easy learning. We also looked at research on how educators might respond to different

levels of adaptation, deal with adversity, and foster resilience to help all students mature in school. Potentially, learning styles may not exist in the way that their proponents describe them, making it impossible to identify and educate students based on them. During the epidemic, students chose to learn online, in flipped courses with project-based activities, as well as in groups.

## 9. Future Research

Future research should consider the perspectives of instructors and other higher-education partners on the most effective learning approaches to use in academic contexts. Future studies should investigate limits and permit activities given that unique perspectives from various areas and ethnicities throughout the world would undoubtedly improve the research. Teachers should be given advice on how to incorporate the learning approach into various areas of learners' learning stages in the future. Future studies may give further insight on how to address the problem in academic settings. Finally, respondents to the questionnaire in this study came from just one institution, and those respondents were largely from South Korea. In future research, we can include more international students and other universities.

**Author Contributions:** Conceptualization, M.U. and I.K.; methodology, M.U.; software, M.U.; validation, M.U. and I.K.; formal analysis, M.U.; investigation, M.U. and I.K.; resources, M.U. and I.K.; data curation, M.U. and I.K.; writing—original draft preparation, M.U. writing—review and editing, M.U. and I.K.; visualization, M.U. and I.K. supervision, I.K.; project administration, M.U.; funding acquisition, I.K. All authors have read and agreed to the published version of the manuscript.

**Funding:** This research received no external funding.

**Institutional Review Board Statement:** Study did not require ethical approval. We conducted our survey under Chonnam National University's Ethics Compliance.

**Informed Consent Statement:** Informed consent was obtained from all persons involved in the survey.

**Data Availability Statement:** Not applicable.

**Conflicts of Interest:** The authors declare no conflict of interest.

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
