# Peer review of "E-Learning: Direct Effect of Student Learning Effectiveness and Engagement through Project-Based Learning, Team Cohesion, and Flipped Learning during the COVID-19 Pandemic"

_sustainability, doi:10.3390/su14031724_

Round 1
Reviewer 1 Report
In my opinion, after the review of the state of knowledge in the Introduction, it would be worth formulating a research problem. The research problem should result from a knowledge review indicating an insufficient state of knowledge in the area in question. Only the formulation of the research problem should be a premise to identify a gap in the current state of knowledge and to present the purpose of the research. Generally, in the case of the formulated research goal, it would be worth pointing to a cognitive (scientific) goal and a utilitarian (useful) goal at the same time. Even if the authors specified the purpose (s) of the research, it is worth defining unequivocally which of them are cognitive and which are utilitarian.
I have read Section 2.1 of Learning Effectiveness and did not learn how to properly understand the word Effectiveness. Much information in this section has been written about Learning and the factors that determine the state of the process. However, I am - not only as a reviewer, but above all as a reader of the article - interested in how to interpret the word Effectiveness, in a given case related to Learning. I think it would be worthwhile here or elsewhere to provide a definition or approach to interpreting the concept of Effectiveness. This concept (Effectiveness) can be interpreted according to the area of ​​scientific considerations (and not only), therefore it certainly also applies to Learning. I would like to know what this interpretation is. Anyway, the authors use the word Learning Effectiveness from the very beginning of the Introduction and theoretically every reader should know what it is about. However, probably not everyone needs to know what's going on. Of course, in Table 1, the Authors cited many publications taking into account Learning Effectiveness, but I would like to know what was the definition formulated by the Authors and used in the presented research.
Why in the text the Authors use the following notation in the text: table 02 (line: 90), table 03 (line: 103) and table 04 (line: 123), if the designation of the tables is: Table 2, Table 3 and Table 4? The same remark also applies to the other tables.
For example, instead of writing "The [65] asserts ..." (line: 167), you might write "Ref. [65] asserts… ”.
If the acronym PBL is used for the first time (in Table 9), it would be worth mentioning its full name right away. The full name of the acronym was only clarified on line 330 and should be given earlier.
If the Authors provide the names of the authors of the publication in Tables 1-18, it would be worth keeping the same rule in all tables. Some tables only list surnames, others give surnames and the initial letter of the first name, and others give the surname and full name. The spelling rules for the authors of the publication need to be corrected.
Data on the statistical package used to analyze the data (line: 436) should be provided in References. Therefore, in point 4, citation [...] must be included, and the details, i.e. company - producer of statistical software, city, country, etc., should be provided in References.
Chapter 4 is extremely short and laconic and therefore needs to be completed in my opinion. Study participants need to be described in more detail. It is worth providing the average age of students, you can also provide the standard deviation (SD), as well as the age range (the age of the youngest and oldest student participating in the study). These are elements of descriptive statistics and it is worth using. An important issue in the description of the research group, i.e. students, is the fields of study they studied. Were they faculties in humanities or natural sciences (life sciences), or maybe medical, economics or music? The description of the research group (students) provides information on the percentage of women and men in the total population of respondents and their countries of origin. In my opinion, this is not complete information. I think that the above-mentioned missing information about the students may have a significant impact on the obtained research results and be useful in a fuller discussion of the research results. Especially if some of the questions in the survey were related to development prospects, which may show significant connections with the field of study (specialization) of the completed studies.
I would ask the Authors to explain what they meant by "Cumulative percent" in Table 20 and how this value was calculated. It can be assumed that every reader should know this, but this assumption may turn out to be wrong.
I do not know how the Authors calculated "a survey return rate" (lines: 435-436). My calculations show (247/250) that it was 98.8%. However, the authors gave the value of 97.8 (line: 436), and in addition without the "%" sign. Please explain.
I would like to ask for information which Coefficients are given in the right column of Table 23; what is the full name of these coefficients?
I would like to ask about the survey information collection technique that was used in the presented research. Was the CAWI (Computer-Assisted Web Interview) method used, or maybe some other method? It is worth writing about it in the article. CAWI is a technique for gathering information in quantitative market and public opinion research, in which the respondent is asked to complete an electronic questionnaire. In what period was the survey conducted? How long was the survey available on the Internet if data was collected in this way? It is worth writing about it in point 4. You can write how many questions were raised in the questionnaire with students in total. It is true that the knowledge about the individual examined features was developed earlier, but in point 4 it can be presented as a number, for the sake of clarity. Was a pilot survey carried out prior to the main questionnaire to check, for example, the understanding of particular issues in the questionnaire by the respondents? The pilot survey allows you to correct some of the questions in the survey. Has such a correction been made?
I would like to ask if in the group of 247 questionnaires all the questionnaires met the required quality / accuracy criteria of the answers? Were there problematic surveys that had to be rejected? It would be worth writing about it in an article.
It is assumed that in a classic scientific article the results of own research are discussed, i.e. confronted with the research results of other authors. I suggest completing this part of the article.
I think that the topic undertaken by the authors may refer to the search for innovative solutions in educating students. A good example in this regard is the publication "The topic of the ideal dairy farm can inspire how to assess knowledge about dairy production processes: A case study with students and their contributions", which is worth quoting in the article in chapter 8: Contribution. In this way, the prospects for further research can be shown. When developing the issue of learning effectiveness in the article, it is worth quoting the material: “The quest for knowledge transfer efficacy: blended teaching, online and in-class, with consideration of learning typologies for non-traditional and traditional students”.
Finally, I would like to ask what was the reason for the share of students from as many as 15 countries and what was the reason for the large disproportion in the number of respondents from different countries?
I would also like to ask what is the reason for the phrase in line 524 (in the Conclusions): "I hope this research ...". After all, there are two Authors and not one Author.
Author Response
Reviewers’ Comments and Authors Response
Response: The authors would like to thank the reviewer for their comments. Care has been taken to improve the work and address their concerns as per the specific comments below.
We have rewrite the manuscript. We have made substantial changes in the part of the paper to address the comments and suggestions.
|
Reviewer Comments |
Reply |
|
In my opinion, after the review of the state of knowledge in the Introduction, it would be worth formulating a research problem. The research problem should result from a knowledge review indicating an insufficient state of knowledge in the area in question. Only the formulation of the research problem should be a premise to identify a gap in the current state of knowledge and to present the purpose of the research. Generally, in the case of the formulated research goal, it would be worth pointing to a cognitive (scientific) goal and a utilitarian (useful) goal at the same time. Even if the authors specified the purpose (s) of the research, it is worth defining unequivocally which of them are cognitive and which are utilitarian. |
Rewrite the Amended “Effective learning, this initiative is based on research by certain teachers and educators who have worked to guarantee that instructors may develop practices in ambitious directions so that all students, especially those who were previously underserved by schools, can enjoy rich learning opportunities. Many concepts, such as "learning for knowledge," have been developed by educational scholars to describe learning and learning approaches that rely on students' understanding. While some of these concepts pertain to slight changes in approach, the majority of them imply that teachers should solicit, ana-lyze, and investigate students' emerging ideas. In such a learning perspective, awareness is of the highest significance; it is actively negotiated by students and instructors, and it evolves as students' ideas arise and mature. The goal of this research is to examine the direct effects of student learning effectiveness and engagement. One major difficulty is determining how students practice and ensure participation and engagement in the light of unique learner qualities and digital interactions [1]. As we begin our creative approach, it is critical to identify different types of learning factors and their relationships to intermediaries. An interdisciplinary study of teacher-student relationships is a healthy connection with benefits for this research.
The predictable outcome will be examined in this study: • How much do, project-based learning, team cohesion, and flipped learning have direct effects on student learning effectiveness, and engagement? • How much do, engagement direct effects on student learning effectiveness.
This arguments describe the student's learning approaches and how they affect learning directions. The constructivist premise that learning is impacted by how an individual participates in academically intentional activities underpins the concept of student learning. However, learning is viewed as a collaborative effort that requires institutions and personnel to provide students with the circumstances, expectations, and opportunities, to participate. Individual learners, are the ultimate participants in engagement dia-logues. Through personal advantages, students may make a significant contribution to the quality of the learning environment, their peers' experiences, and the greater campus community. Unfortunately, these potentially beneficial benefits are not always fully realized. Fear is playing an important part in the current environment, which is distracting students from studying”. |
|
I have read Section 2.1 of Learning Effectiveness and did not learn how to properly understand the word Effectiveness. Much information in this section has been written about Learning and the factors that determine the state of the process. However, I am - not only as a reviewer, but above all as a reader of the article - interested in how to interpret the word Effectiveness, in a given case related to Learning. I think it would be worthwhile here or elsewhere to provide a definition or approach to interpreting the concept of Effectiveness. This concept (Effectiveness) can be interpreted according to the area of scientific considerations (and not only), therefore it certainly also applies to Learning. I would like to know what this interpretation is. Anyway, the authors use the word Learning Effectiveness from the very beginning of the Introduction and theoretically every reader should know what it is about. However, probably not everyone needs to know what's going on. Of course, in Table 1, the Authors cited many publications taking into account Learning Effectiveness, but I would like to know what was the definition formulated by the Authors and used in the presented research. |
Amended “Learning effectiveness is now focusing on student learning; it is quite an active process, formal or informal. It is a static group of reality, information, and skills. Learning is changing, in effect, awareness, behaviors, abilities, morals, and preferences. Learning effectiveness refers to the entire process through which students participate in a high-quality learning opportunity. Within a system that fosters student growth, a quality education involves quantifiable results related to well-defined learning standards. Specific design features such as cognitive, instructional, and social representation in the e-learning network have been used to assess learning efficacy. Through thought and research, these design components aim to cultivate and facilitate higher-level thinking skills. As a question that learning effectiveness addresses, "Do I understand more than I knew before? How will this new knowledge help me?" Effective learning is the capacity of a learner to explicitly indicate what they have learnt through quantitative measures. Learning begins with the formulation of particular learning outcomes that are aligned with the courses or program's general goals and objectives”. |
|
Why in the text the Authors use the following notation in the text: table 02 (line: 90), table 03 (line: 103) and table 04 (line: 123), if the designation of the tables is: Table 2, Table 3 and Table 4? The same remark also applies to the other tables. |
This has now been corrected as suggested. |
|
For example, instead of writing "The [65] asserts ..." (line: 167), you might write "Ref. [65] asserts… ”. |
This has now been corrected as suggested. |
|
If the acronym PBL is used for the first time (in Table 9), it would be worth mentioning its full name right away. The full name of the acronym was only clarified on line 330 and should be given earlier. |
This has now been corrected as suggested. |
|
If the Authors provide the names of the authors of the publication in Tables 1-18, it would be worth keeping the same rule in all tables. Some tables only list surnames, others give surnames and the initial letter of the first name, and others give the surname and full name. The spelling rules for the authors of the publication need to be corrected. |
This has now been corrected as suggested. |
|
Data on the statistical package used to analyze the data (line: 436) should be provided in References. Therefore, in point 4, citation [...] must be included, and the details, i.e. company - producer of statistical software, city, country, etc., should be provided in References.
|
Amended “In this research, the supervisor reviewed the survey questionnaire, determining the information needed, deciding on a question topic, and developing question content. The questionnaire's criterion was that respondent understood the question's objective. Before starting the survey, each class had an introduction of questionnaire and purpose of the survey. A questionnaire is a type of quantitative research tool that consists of many questions that are used to collect data from respondents. 250 survey copies was distributed, three questionnaires were discarded because respondents provided incomplete survey information. Two hundred forty-seven effective samples of the questionnaire were complete returned, with a survey return rate of 98.8 %. In-person feedback was used to acquire data for the survey. The Statistical Product software package SPSS 25 was employed to calculate the statistics with respect to data processing. A data-gathering survey was conducted with a specific set of business management students. The survey was done among undergraduate and graduate business management students at Chonnam National University in various program/courses in the school of business. Participants in the poll come from different countries and are enrolled in courses at Chonnam National University”.
|
|
Chapter 4 is extremely short and laconic and therefore needs to be completed in my opinion. Study participants need to be described in more detail. It is worth providing the average age of students, you can also provide the standard deviation (SD), as well as the age range (the age of the youngest and oldest student participating in the study). These are elements of descriptive statistics and it is worth using. An important issue in the description of the research group, i.e. students, is the fields of study they studied. Were they faculties in humanities or natural sciences (life sciences), or maybe medical, economics or music? The description of the research group (students) provides information on the percentage of women and men in the total population of respondents and their countries of origin. In my opinion, this is not complete information. I think that the above-mentioned missing information about the students may have a significant impact on the obtained research results and be useful in a fuller discussion of the research results. Especially if some of the questions in the survey were related to development prospects, which may show significant connections with the field of study (specialization) of the completed studies. |
A data-gathering survey was conducted with a specific set of pupils. The survey was done among undergraduate and graduate business management students at Chonnam National University in various courses in the school of business. Participants in the poll come from different countries and are enrolled in courses at Chonnam National University. The initial step in our analysis was to compile descriptive statistics for the variables in our research. Project based learning (PBL) ranged from 1.00 to 5.00, with a mean of 3.87 and a standard deviation of.916, according to the descriptive statistics for our sample. With varying means and standard deviations, TC, flipped learning, engagement, and learning effectiveness ranged from 1.00 to 5.00. (See Table 4). In Table 3, most of the country-wise participants from South Korea, Uzbekistan, China, Pakistan, and other neighboring countries in Asia, Europe, and Africa. In the survey, we asked participants only a few personal questions, i.e., name, gender, e-mail, and nationality. It is a little too personal question, the age was not mentioned in the survey. We believed that the majority of the students were between the ages of 21 and 32.
|
|
I would ask the Authors to explain what they meant by "Cumulative percent" in Table 20 and how this value was calculated. It can be assumed that every reader should know this, but this assumption may turn out to be wrong. |
This has now been corrected. On the paper wrong table was used. |
|
I do not know how the Authors calculated "a survey return rate" (lines: 435-436). My calculations show (247/250) that it was 98.8%. However, the authors gave the value of 97.8 (line: 436), and in addition without the "%" sign. Please explain. |
This has now been corrected as suggested. It was typing mistake. |
|
I would like to ask for information which Coefficients are given in the right column of Table 23; what is the full name of these coefficients? |
The full name of coefficients is coefficients” . It was taken from hierarchal linear regression.
|
|
I would like to ask about the survey information collection technique that was used in the presented research. Was the CAWI (Computer-Assisted Web Interview) method used, or maybe some other method? It is worth writing about it in the article. CAWI is a technique for gathering information in quantitative market and public opinion research, in which the respondent is asked to complete an electronic questionnaire. In what period was the survey conducted? How long was the survey available on the Internet if data was collected in this way? It is worth writing about it in point 4. You can write how many questions were raised in the questionnaire with students in total. It is true that the knowledge about the individual examined features was developed earlier, but in point 4 it can be presented as a number, for the sake of clarity. Was a pilot survey carried out prior to the main questionnaire to check, for example, the understanding of particular issues in the questionnaire by the respondents? The pilot survey allows you to correct some of the questions in the survey. Has such a correction been made? |
A questionnaire is a type of quantitative research tool that consists of many questions that are used to collect data from respondents. The respondents are a sample of the population that takes part in the data-gathering survey. In-person feedback was used to acquire data for the survey. A data-gathering survey was conducted with a specific set of pupils. The survey was done among undergraduate and graduate business management students at Chonnam National University in various courses in the school of business. Participants in the poll come from different countries and are enrolled in courses at Chonnam National University. The supervisor reviewed the survey questionnaire, determining the information needed, deciding on a question topic, and developing question content.
|
|
I would like to ask if in the group of 247 questionnaires all the questionnaires met the required quality / accuracy criteria of the answers? Were there problematic surveys that had to be rejected? It would be worth writing about it in an article. |
The questionnaire has a total of 68 questions for the students to answer. The questionnaire's criterion was that the respondent understood the question's objective. Before starting the survey, each class had an introduction and purpose for the survey. Three questionnaires were discarded because respondents provided incomplete survey information. |
|
It is assumed that in a classic scientific article the results of own research are discussed, i.e. confronted with the research results of other authors. I suggest completing this part of the article. |
Rewrite “In our research, we discovered that project-based learning (PBL), team cohesiveness, and flipped learning had direct effects on student learning effectiveness and engagement. Not only is there a statistically significant positive relationship, but there is also a statistically negative relationship. Research showed that project-based learning (PBL) had a beneficial impact on student learning effectiveness and engagement as predicted. Several studies [39, 41, 46, and 49] have mentioned research variables in project-based learning (PBL).The results were also compared to other studies that indicated the parameters had a substantial impact on student learning effectiveness [41], [44]. On the other hand, the findings, revealed that team cohesion plays a dominant role in the student's learning effectiveness and engagement. Despite the fact that research variables were counted in receptive studies [51], [54], and [55], the results were found to be significantly similar to those of studies that claimed the factors were important in influencing student learning effectiveness [54], [55], and [57]. Another noteworthy observation is that while flipped learning had a statistically significant beneficial effect on student learning effectiveness, it also had an unanticipated negative effect on engagement. Flipped learning has little contact with student learning studies that claim [21], [66], [70], and [73]. The results were also compared to those of previous studies that affirmed the importance of various elements in determining student learning effectiveness [65], [69], and [70]. Finally, while engagement has a direct impact on student learning effectiveness, it is not well substantiated.
In the COVID situation, most of the students enrolled in online coursework. The impact of the study on university-level local and international students can be learnt from future research and advancement. This study involved undergraduates with appropriate student engagement, their own class or group experiences, and observation. Students prefer active learning environments over passive classroom settings, and they expect the same in their classrooms because they live in such an engaging society. Today's students want more opportunities for creativity and collaboration, which virtual education environments may deliver through a range of teaching methods that can be accessed at any time and from any location [55], previous studies have shown that students' participation helps them to achieve academic improvement and create a more motivating learning environment”. |
|
I think that the topic undertaken by the authors may refer to the search for innovative solutions in educating students. A good example in this regard is the publication "The topic of the ideal dairy farm can inspire how to assess knowledge about dairy production processes: A case study with students and their contributions", which is worth quoting in the article in chapter 8: Contribution. In this way, the prospects for further research can be shown. When developing the issue of learning effectiveness in the article, it is worth quoting the material: “The quest for knowledge transfer efficacy: blended teaching, online and in-class, with consideration of learning typologies for non-traditional and traditional students”. |
Amended “Furthermore, because university education has substantial socioeconomic repercussions [98], it's a great example of a "case study including students and their contributions" for increasing learning efficacy. The aim of the research was to look at the idea of an ideal cow ranch from the views of two different groups. A group of young people were motivated to propose a research method that would encourage students to reflect on and assess their own knowledge in comparison to that of others. When learning patterns for non-traditional and traditional students were taken into account, only a tiny fraction of students were employing terminology that showed a logical approach to dairy farming, such as ethical concerns, liberty, sustainability, wildlife distress, and organic dairy production. According to the findings of the case study, a curriculum that will help young people become more aware of current dairy production concerns should be developed. To learn more successfully during the epidemic, training and online classes might be a preferable alternative. As a result, governments are concerned about ensuring a continuous flow of students enrolling in higher education”.
|
|
Finally, I would like to ask what was the reason for the share of students from as many as 15 countries and what was the reason for the large disproportion in the number of respondents from different countries? |
The school of business at Chonnam National University has a low number of international students. The percentage of international students in the business school is minimal. |
|
I would also like to ask what is the reason for the phrase in line 524 (in the Conclusions): "I hope this research ...”. After all, there are two Authors and not one Author. |
This has now been corrected as suggested. |

Reviewer 2 Report
Dear Authors,
Please, take into account the following issues:
- the authors should carefully read the INSTRUCTIONS FOR AUTHORS at Sustainability | Instructions for Authors (mdpi.com).
- The section ABSTRACT should be re-organized. The aims of the paper should be placed before the methods. It would be better to re-formulate the following phrases: "The primary goal of this study was to figure out the direct effect of student learning effectiveness. The secondary goal of this study was to figure out the indirect effect of student learning effectiveness. The secondary goal of this study was to figure out the indirect effect of student learning effectiveness." (lines 12-14) Summarize the article's main findings and present the main conclusions- it is too short.
- Some phrases are rather ambiguous in the INTRODUCTION section- e.g., lines 21-22. Also, the aims of the paper, stated in the previous section, are not the same with "The purpose of this study is to look at the direct and indirect effects of student learning effectiveness on university students' output" (lines 24-25) and "The goal of this research is to calculate the ability of students to learn with direct and indirect approaches." (lines 53-54). It would be better to re-formulate the two phrases from line 30 to 33- only one word is different. This section should briefly place the study in a broad context and highlight its importance. Also, it should clearly present the specific hypotheses to be tested and highlight the main conclusions.
- The section THEORETICAL BACKGROUND is too long and should be restructured. It has to deal with the keywords of the paper: define them, present them, if possible, in connection etc. It would be better to summarize the key findings. Is it necessary to present so many concepts/terms?
- It would be better to move the theoretical framework of the model (summary) from section 6 to section 5.
- The authors should discuss the results and how they can be interpreted in perspective of previous studies and of the working hypotheses. The findings and their implications should be discussed in the broadest context possible and limitations of the work highlighted. Future research directions may also be mentioned.
Good luck!
Author Response
Reviewers’ Comments and Authors Response
Response: The authors would like to thank the Reviewer for their comments. Care has been taken to improve the work and address their concerns as per the specific comments below.
We have rewritten the manuscript. We have made substantial changes in the part of the paper to address the comments and suggestions.
|
Reviewer Comments |
Reply |
|
The authors should carefully read the INSTRUCTIONS FOR AUTHORS at Sustainability | Instructions for Authors (mdpi.com). |
Noted |
|
The section ABSTRACT should be re-organized. The aims of the paper should be placed before the methods. It would be better to re-formulate the following phrases: "The primary goal of this study was to figure out the direct effect of student learning effectiveness. The secondary goal of this study was to figure out the indirect effect of student learning effectiveness. The secondary goal of this study was to figure out the indirect effect of student learning effectiveness." (lines 12-14) Summarize the article's main findings and present the main conclusions- it is too short. |
Amended “The cross-impact of project-based learning, team cohesion, and flipped learning was investigated by examining their direct effects on student learning effectiveness, engagement, and engagement effects on learning effectiveness. The results of hypotheses testing using hierarchical regression analysis with SPSS-25 statistical packages for data analysis. The research model was empirically verified with quantitative data have collected from 247 graduate/undergraduate business students based on their own experiences, observations, and engagement. Analysis found that project based leaning (PBL), team cohesion increased positive direct effects in student learning effectiveness and engagement. On other hand flipped learning increased positive direct effects in student learning effectiveness, but negative effects in engagement. Engagement had a positive direct effects on student learning effectiveness. This study's conclusions were intended to inform practice in terms of increasing retention and enhancing teaching and student learning quality”. ”. |
|
Some phrases are rather ambiguous in the INTRODUCTION section- e.g., lines 21-22. Also, the aims of the paper, stated in the previous section, are not the same with "The purpose of this study is to look at the direct and indirect effects of student learning effectiveness on university students' output" (lines 24-25) and "The goal of this research is to calculate the ability of students to learn with direct and indirect approaches." (lines 53-54). It would be better to re-formulate the two phrases from line 30 to 33- only one word is different. This section should briefly place the study in a broad context and highlight its importance. Also, it should clearly present the specific hypotheses to be tested and highlight the main conclusions. |
Rewrite “Effective learning, this initiative is based on research by certain teachers and educators who have worked to guarantee that instructors may develop practices in ambitious directions so that all students, especially those who were previously underserved by schools, can enjoy rich learning opportunities. Many concepts, such as "learning for knowledge," have been developed by educational scholars to describe learning and learning approaches that rely on students' understanding. While some of these concepts pertain to slight changes in approach, the majority of them imply that teachers should solicit, analyze, and investigate students' emerging ideas. In such a learning perspective, awareness is of the highest significance; it is actively negotiated by students and instructors, and it evolves as students' ideas arise and mature. The goal of this research is to examine the direct effects of student learning effectiveness and engagement.”
|
|
The section THEORETICAL BACKGROUND is too long and should be restructured. It has to deal with the keywords of the paper: define them, present them, if possible, in connection etc. It would be better to summarize the key findings. Is it necessary to present so many concepts/terms? |
Rewrite and re arranged the theoretical background. |
|
It would be better to move the theoretical framework of the model (summary) from section 6 to section 5. |
This now been corrected as suggested. |
|
The authors should discuss the results and how they can be interpreted in perspective of previous studies and of the working hypotheses. The findings and their implications should be discussed in the broadest context possible and limitations of the work highlighted. Future research directions may also be mentioned. |
Rewrite Practical Implications and Limitation
“In this study, it was a beneficial tool in identifying how to measure the effectiveness of learning studies. Methodological research must continue to account for measurement error when studying self-performance and can add other aspects to the mix, such as game-based learning and cooperative learning. Students who are unfamiliar, perplexed, unwelcoming, or unsupportive in the classroom will struggle. More institutions, other relevant aspects that influence the effectiveness of student learning can be included. While the [99] authors contend that the earlier study's reaction to the input explanation was quite low, and while idlers are common in empirical investigation and may reduce the comparability of assumptions, a new study is needed to determine the best framework for analyzing student learning effectiveness. This study examines the consequences of a developing consensus on student learning and effectiveness for students and classroom methods, as well as a modern mix of elements that connect digital learning. As a result of their learning experiences, students experience anxiety, frustration, social isolation, sadness, and a lack of self-confidence. Foreign students studying abroad, in particular, experience language barriers, cultural norms, value issues and different education system. Study would assist learners in providing comfort, healthy growth, and easy learning. We also looked at research on how educators might respond to different levels of adaptation, deal with adversity, and foster resilience in order to help all students mature in school. It's possible that learning styles don't exist in the way that their proponents describe them, making it impossible to identify and educate students based on them. During the epidemic, students chose to learn online, in flipped courses, with project-based activities, and in groups”.
Future Research “Future research should consider the perspectives of instructors and other higher education partners on the most effective learning approaches to use in academic contexts. Future studies should investigate limits and permit activities given that unique perspectives from various areas and ethnicities throughout the world would undoubtedly improve the research. Teachers should be provided advice on how to incorporate the learning approach into various areas of learners' learning stages in the future. Future studies may give further insight on how to address the problem in academic settings. Finally, respondents to the questionnaire in this study came from just one institution, and those respondents are largely from South Korea. In future research, we can include more international students and include other universities”.
|

Round 2
Reviewer 1 Report
Thank you for preparing the answers to my questions and suggestions in the review, as well as the changes and additions made to the article.
Author Response
Response to Reviewer Comment
The authors are very grateful for the reviewers’ kind and constructive comments on the paper (sustainability-1511636). We did our best to incorporate all of the reviewer comments in the revised paper. The authors hope this clears out the questions raised by reviewers. Here is the copy of the decision letter received
|
Reviewer Comment |
Reply |
|
Thank you for preparing the answers to my questions and suggestions in the review, as well as the changes and additions made to the article.
|
The authors are grateful for the kind and valuable comments of the respected reviewer that has significantly improved the quality of the manuscript. In addition, minor spell check is done.
We highlight re-structure paragraphs and spelling checks in a yellow mark.
|

Reviewer 2 Report
Dear Authors,
You have made substantial improvements. However, you should be more specific when you present the aims of the paper. Also, English language and style are fine/minor spell check required.
Good luck!
Author Response
Response to Reviewer Comment
The authors are very grateful for the reviewer's kind and constructive comments on the paper (sustainability-1511636). We did our best to incorporate all of the reviewers’ comments in the revised paper. The authors hope this clears out the questions raised by reviewers. Here is the copy of the decision letter received
|
Reviewer Comment |
Reply |
|
You have made substantial improvements. However, you should be more specific when you present the aims of the paper. Also, English language and style are fine/minor spell check required.
Good Luck!
|
The authors are thankful for the thoughtful and helpful feedback provided by the respected reviewer. The whole manuscript is proofread thoroughly with respect to the aims of the research and highlighted with yellow in the introduction of the revised paper (as below). In addition, minor spell check is done.
We highlight re-structure paragraphs and spelling checks in a yellow mark.
|
